# SpaceSet: A Large-scale Realistic Space-based Image Dataset for Space Situational Awareness

## Abstract

Space situational awareness (SSA) plays an imperative role in maintaining safe space operations, especially given the increasingly congested space traffic around the Earth. Space-based SSA offers a flexible and lightweight solution compared to traditional ground-based SSA. With advanced machine learning approaches, space-based SSA can extract features from high-resolution images in space to detect and track resident space objects (RSOs). However, existing spacecraft image datasets, such as SPARK, fall short of providing realistic camera observations, rendering the derived algorithms unsuitable for real SSA systems. In this research, we introduce SpaceSet, a large-scale realistic space-based image dataset for SSA. We consider accurate space orbit dynamics and a physical camera model with various noise distributions, generating images at the photon level. To extend the available observation window, four overlapping cameras are simulated with a fixed rotation angle. SpaceSet includes images of RSOs observed from $19km$ to $63,000km$, captured by a tracker operating in LEO, MEO, and GEO orbits over a period of $5,000$ seconds. Each image has a resolution of $4418 \times 4418$ pixels, providing detailed features for developing advanced SSA approaches. We split the dataset into three subsets: SpaceSet-100, SpaceSet-5000, and SpaceSet-full, catering to various image processing applications. The SpaceSet-full corpus includes a comprehensive data-loader with $781.5GB$ of images and $25.9MB$ of ground truth labels. We also benchmark detection and tracking algorithms on the SpaceSet-100 dataset using a specified splitting method to accelerate the training process.

## 1 Introduction

Space Situational Awareness (SSA) Wang et al. (2022) plays a crucial role in ensuring the safety of space assets by providing real-time information perception and risk evaluation for space operations, such as spacecraft navigation Hein (2020) and debris mitigation Usovik (2023). Conventional SSA systems, like those used by the Japanese Space Agency (JAXA) Harris et al. (2021), rely on observing resident space objects (RSOs) and determining their orbits using ground-based facilities equipped with large telescopes and radars. These systems necessitate extensive site areas, high costs, and specific geographical locations. Given the computational limitations of satellites, current space-based SSA systems, which involve complex numerical calculations, typically depend on the space-ground network for data processing and information fusion. This reliance results in substantial communication loads and delays.

With the advancements in artificial intelligence (AI) and high-performance edge computing, an on-board vision-based SSA system presents a more flexible and lightweight alternative to traditional ground-based SSA for RSO detection and tracking. One of the primary challenges in SSA is providing accurate position and orientation vectors (observations) of targets to determine their orbits. Methods such as Gauss's method Vallado (2001), which requires at least three observations for preliminary orbit determination, and Lambert's method Engels & Junkins (1981), which needs only two position vectors with temporal information, are used for this purpose. Essentially, increasing the number of observations enhances the accuracy of orbit determination, highlighting the importance of the object detection and tracking (ODT) component in SSA. To develop precise and practical ODT

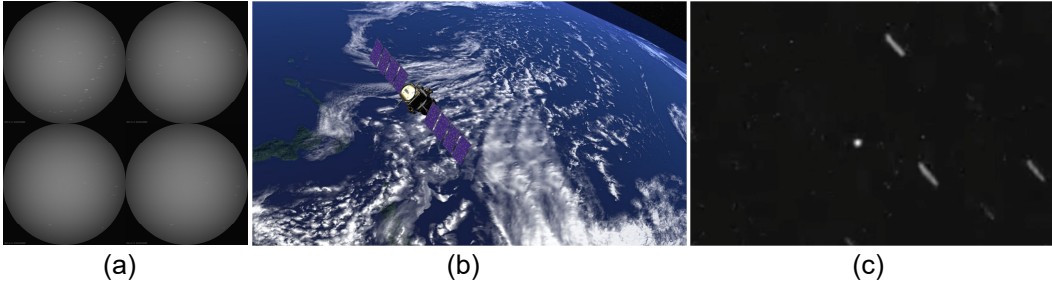

(a)  (b)  (c)

Figure 1: Comparison of our SpaceSet images with SPARK images Musallam et al. (2021a) and real-life observed images. (a) SpaceSet images at timestamp 0 (four cameras from left top to right bottom), which show the realistic exposure with noise distribution; (b) A simulated spacecraft image from SPARK; (c) The real-life space observation image from the telescope and sensor network (EGTN[2]). The similar streaks due to the exposure of fast-moving RSOs and the hot pixels induced by the noises in (a) and (c) demonstrate the realistic images in our SpaceSet dataset.

algorithms, extensive high-resolution space imagery is essential. However, most existing spacecraft image generation techniques Musallam et al. (2021a) rely on high-fidelity simulators that ignore space camera models and the cosmic background, resulting in unrealistic images (See Figure 1) and ODT algorithms unsuitable for real SSA systems.

In this work, we present SpaceSet, a large-scale realistic space-based image dataset for SSA. This dataset considers accurate space orbit dynamics and a physical camera model with various noise distributions, generating images at the photon level. To extend the observation window, we simulate four overlapping cameras with a fixed rotation angle. SpaceSet comprises images of RSOs observed from distances ranging from $19\,\text{km}$ to $63,000\,\text{km}$, captured by a tracker operating in Low Earth Orbit (LEO), Medium Earth Orbit (MEO), and Geostationary Orbit (GEO) over $5,000$ seconds. Each image boasts a resolution of $4418 \times 4418$ pixels, providing detailed features for the development of advanced SSA approaches. We have divided the dataset into three subsets: SpaceSet-100, SpaceSet-5000, and SpaceSet-full, catering to various image processing applications. The SpaceSet-full corpus includes a comprehensive dataloader with $781.5\,\text{GB}$ of images and $25.9\,\text{MB}$ of ground truth labels. To the best of our knowledge, SpaceSet is the first image dataset to offer four-camera observations with realistic image generation from space for space object detection and tracking. The key contributions and features of this dataset are summarized as follows:

**Realistic Image Generation**: Incorporating accurate space orbit dynamics and a physical camera model with various noise distributions to produce photon-level realistic space images.

**Multiple Camera Observations**: Simulating four overlapping cameras with fixed rotation angles to extend the observation window.

**Large Range Tracker Observation**: Covering RSO images observed from $19\,\text{km}$ to $63,000\,\text{km}$ for the tracker operating in LEO, MEO, and GEO orbits.

**Automated Label Generation with Bearing Angle**: Providing accurate ground truth labels with bearing angle information generated by the simulator through an automated transformation and annotation process.

**Extensive Benchmarks**: Benchmarking the dataset using SOTA algorithms, including YOLOv5, YOLOv8, YOLOv10, DINO, etc., on SpaceSet-100 with a specified splitting method to expedite the training process. Additionally, various object tracking methods are compared on SpaceSet-100 to explore its applications.

## 2 RELATED DATASET WORK

Publicly available image datasets for space object imagery are predominantly ground-based, such as SatNet and SatSim Fletcher et al. (2019). Existing space-based image datasets, such as BUAA-

SID-POSE 1.0 Qiao et al. (2022), SPEED Kisantal et al. (2020), SPEED+ Park et al. (2023), and URSO Proença & Gao (2020), primarily emphasize spacecraft pose estimation Pauly et al. (2023). These datasets typically feature a limited number of RSOs in the images and lack comprehensive annotations such as bounding boxes, which are essential for broader SSA applications beyond pose estimation.

Since space-borne real data is often challenging and expensive to acquire, simulated datasets have become the predominant approach for developing methods for SSA tasks. BUAA-SID 1.0 Zhang et al. (2010) features various satellite models created using 3dsMax but lacks simulation of the space environment. The SPARK Musallam et al. (2021a) dataset includes simulated models of different satellites and space debris but lacks realistic camera observations. An annotated dataset derived from the Resident Space Object Near-Space Astrometric Research (RSONAR) mission is provided by Suthakar et al. (2023), which collected data using a low-resolution, wide-field-of-view imager on a stratospheric balloon. Additionally, some datasets have been generated by researchers to simulate space conditions and RSOs, facilitating algorithm development and testing Tang et al. (2023); Chen et al. (2023); Shen et al. (2024). However, these datasets are often inaccessible and lack comprehensive reality analysis. Table 1 provides a summary of statistics for existing space-based RSO detection image datasets as well as our SpaceSet dataset. SpaceSet captures more RSOs in the images and has a higher resolution compared to prior datasets.

Table 1: Comparisons of SpaceSet with existing datasets.

| Dataset | #Images | #Objects | Resolution | Object/Image | Public? |
|---|---|---|---|---|---|
| BUAA-SID-share 1.0 Zhang et al. (2010) | 9.2k | 20 | 320×240 | single | yes |
| SPARK Musallam et al. (2021a) | 30k | 11 | 1440× 1080 | single | request |
| RSONAR Suthakar et al. (2023) | 429 | 3 | 1024× 1024 | multiple | no |
| **SpaceSet-100** | 100 | 56 | 4418 × 4418 | multiple | yes |
| **SpaceSet-5000** | 5k | 414 | 4418 × 4418 | multiple | yes |
| **SpaceSet-full** | 20k | 673 | 4418 × 4418 | multiple | yes |

## 3 DATA CURATION PROCESS

### 3.1 DATA GENERATION

The SpaceSet dataset is collected from a real-time high-fidelity simulator based on precise space orbit dynamics and physical camera models. Since the space-based observer operates at an altitude of 500 km, the effects of the atmosphere and related noise are not included in the modeling process. The space environment model simulates a catalog of RSOs in orbit around the Earth, along with other celestial bodies in the sky. The RSO simulation is based on the United States 18th SDS Space Catalog [3], which is fetched in Two-Line Elements (TLEs) format for the desired simulation epoch and propagated using an SGP4 propagator. The propagator provides the positions and velocity vectors of all objects in the TEME coordinate system, which is used to populate the 3D environment.

The modeling of environmental noise expected for a sensor is also incorporated. Under favorable imaging conditions, the sensor's payload is oriented away from the sun and perpendicular to the orbit, allowing the primary background noise source to be the Earth's limb—the bright edge of the Earth's horizon. This background noise is modeled using data from the Hubble Space Telescope [4] and the NEOSSAT mission Thorsteinson (2018). The camera captures a circular image on the image plane, and the detector on the focal plane records the digital image.

To detect which RSOs crossing the field of view (FOV) can be identified by the sensor, a photometric detection model is applied. The sensor detects an object only when its signal-to-noise ratio (SNR) exceeds a specified threshold, typically set at 5. The received signal is calculated assuming 100% diffuse reflection, where the fraction of incident sunlight reflected to the sensor is given by:

---

[2]https://exoanalytic.com/space-domain-awareness

[3]http://space-track.org/

[4]https://hst-docs.stsci.edu/stisihb/chapter-6-exposure-time-calculations/6-5-detector-and-sky-backgrounds

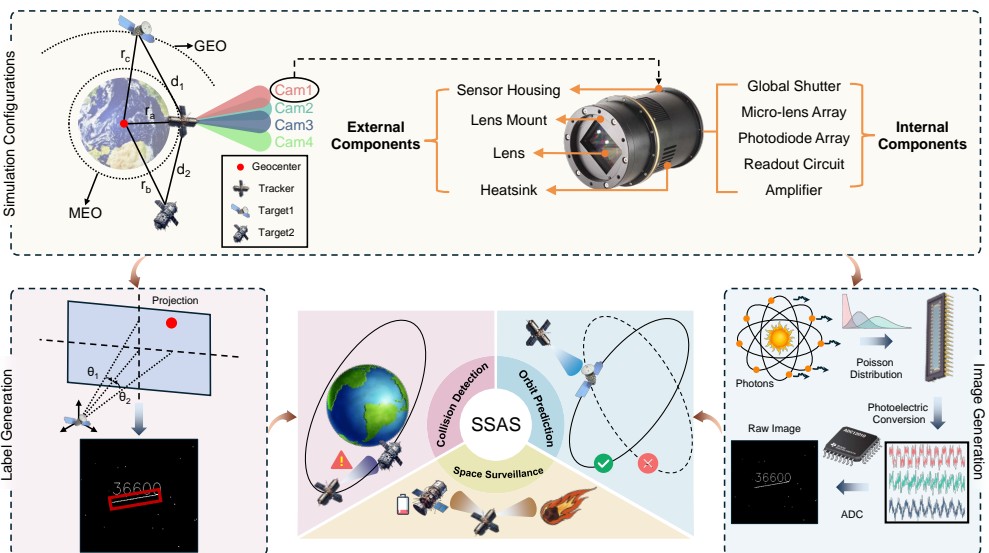

Figure 2: Overall framework of the data curation process.

$$\text{Reflection Factor} = \frac{2\mu r^2}{3\pi R^2} \times (\sin\phi + (\pi - \phi)\cos\phi), \tag{1}$$

where $\mu$ is the object's reflectivity, $r$ is the object's radius, $R$ is the distance between the object and the telescope, and $\phi$ is the phase angle of reflection. The solar flux, considered as the solar constant at 1 AU from the sun, is assumed to be uniform across all objects, as the variation in distance from the sun is negligible. The brightness data from the star catalog is used to determine the signal level from stars, which is distributed across a Gaussian spot formed on the detector.

The model for projecting star/object positions onto the image is based on a pinhole camera model, supplemented by a compound distortion model (radial and tangential), similar to the Brown-Conrady model Zhang (2000). To calculate the noise electrons, three noise sources are modeled: shot noise with a random distribution, sensor noise (e.g., dark current) modeled with a Poisson distribution, and read noise modeled with a normal distribution. Additionally, background noise is modeled with a Poisson distribution. Relative illumination is implemented as a quadratic function to account for roll-off and vignetting effects in the image. These signal and noise models provide the number of photo-electrons collected by each pixel on the sensor, which are then converted into 16-bit digital values (0-65535). Note that four cameras (60°, 75°, 90° and 105° azimuth angle for Cam1 to Cam4, respectively) are adopted to generate the images simultaneously.

The overall framework for the data curation process is illustrated in Figure 2. The datasets generated by the simulator are in the forms of images (*TIFF format*) and a set of metadata (*CSV format*). All *state information* (including position, velocity and attitude) of RSO is propagated with the public in-catalog TLEs. The data collection is free of any ethical issue or participation risk. The space orbit propagation program is developed based on the Standards of Fundamental Astronomy (SOFA) Board package and SGP4 model Vallado & Crawford (2008). With camera specifications such as lens parameters, sensor parameters, and camera pointing direction angles (elevation and azimuth angles in RSW coordinate frame), the physical camera model can generate *pixel values* of images at each timestep based on the aforementioned relative illumination and noise distributions. The physical camera model used for generating realistic space-based images includes several key components, including the pinhole camera model, lens distortion, and noise modeling. Each 3D point $\mathbf{X}$ in the space is projected onto the image plane using the pinhole camera model, then distorted based on the lens distortion model, and finally, various noise distributions as mentioned are added to simulate the physical conditions of space imaging. For instance, we model the noise as a combination of Poisson noise (sensor noise) and Gaussian noise (read noise):

$$I_{noisy}(u, v) = \text{Poisson}(I(u, v)) + \mathcal{N}(0, \sigma^2) \tag{2}$$

where $I(u, v)$ is the intensity value at pixel $(u, v)$, $\text{Poisson}(I(u, v))$ represents the Poisson noise, and $\mathcal{N}(0, \sigma^2)$ represents the Gaussian noise with mean 0 and variance $\sigma^2$. The exposure time (*1 second*) is reflected in the image generation as we overlap the images over the exposure time into one image. The ground truth bounding box is calculated with the bearing angles ($\theta_1$ and $\theta_2$) of a target with respect to the tracker as defined in the Figure 2. By selecting the starting time of simulation (*YYYY-MM-DD HH:MM:SS* in UTC) and simulation duration, we can generate desired images and metadata over a certain period. The specified simulation time is from *2023-01-01 0:00:00* to *2023-01-01 1:23:20* with 1 second time difference for the successive images (this time difference is the exposure time and optimized for object detection).

## 3.2 Dataset Validity and Uniqueness

Currently, there are fewer than six datasets available in this field, and they are all based on simulations, as NASA's database is not publicly accessible. The dataset presented in this work is the first large-scale, realistic, space-based image dataset at the photon level, aiming to bridge the gap between simulated and real-world data. Most existing datasets, such as BUAA-SID-share 1.0 Zhang et al. (2010), SPARK Musallam et al. (2021b), and the Space Target Dataset Zhang et al. (2022c), are primarily generated for satellite pose estimation and space target classification in ideal simulation conditions. These datasets focus on capturing targets from close distances and multiple angles to emphasize single-target characteristics. In contrast, the presented dataset captures targets at various distances based on realistic space-based camera observations. We have compared our images with the real observed images with a ground-based telescope in Appendix A.6, where the starfield and captured positions at various timestamps are first compared to show the accuracy of the simulator. These imaging results then clearly validate the realistic characteristics of our simulated images.

## 3.3 Data Annotation

All images in the SpaceSet dataset are annotated with classes indicating LEO, MEO, and GEO (low, medium, and high accordingly), as well as 2D bounding boxes for the labeled parts (see Figure 3). To ensure high-quality annotations, all classes and bounding box information is automatically derived from the orbital and positional information of space objects, rather than being manually labeled. All the orbital and positional information of these space objects is generated from the aforementioned orbit propagation simulator containing a semi-major axis (SMA) and two bearing angles.

**Classes Annotation**: SMA, a key parameter for describing orbital ellipses, determines the size and shape of the orbit. Targets are classified as LEO (SMA $\leq 8413km$), MEO ($8413km <$ SMA $\leq 42240km$) , and GEO (SMA $> 42240km$) based on their SMA.

**Bounding Boxes Annotation**: Bounding boxes are derived from two bearing angles $\theta_1$ and $\theta_2$ of space objects. The bearing angle information is defined in the camera frame, while the pixel position is defined in the pixel coordinate system (origin at the upper left, x-axis to the right, y-axis downward). The transformation from bearing angles to pixel positions is given by the following equations:

$$x_{\text{pixel}} = \left( \tan(\theta_1) \cdot \frac{\text{focal\_length}}{\text{H\_number}} + 0.5 \right) \cdot \text{width} \tag{3}$$

$$y_{\text{pixel}} = \left( \tan(\theta_2) \cdot \frac{\text{focal\_length}}{\text{H\_number}} - 0.5 \right) \cdot (-\text{width}) \tag{4}$$

The transformations provide the position of the space object at a specific moment in the image. Since each image has a one-second exposure time, the objects appear as a path showing their movement during that second. To find the bounding box, we calculate the pixel coordinates of the space object at the start and end of the exposure. These two points form the diagonal corners of the bounding box, which helps us accurately determine the size and location of each bounding box.

## 3.4 Image Slicing

The original image size is 4418×4418 far beyond the processing capabilities of YOLO and most GPUs. To address the issue of large image dimensions, the images are initially sliced into smaller

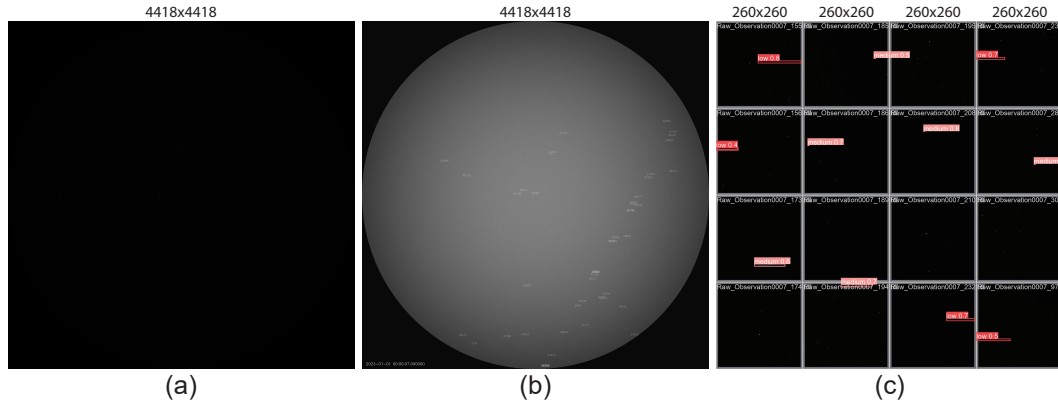

Figure 3: The image sample from the SpaceSet dataset. (a) The original image generated from the simulator with 1 second exposure time; (b) The post-processed image to show the observed image and RSOs with reference IDs; (c) The sliced image batch and annotations.

sections measuring 260×260 pixels each (see Figure 3 (c)). Multiple split sizes were tested, and 260x260 was selected as the optimal size based on experimental results. This slicing process includes an adjustable overlap in both horizontal and vertical directions, serving as a strategy for data augmentation. The last slices are aligned to the image's edge and then cut to the predefined size. Additionally, the annotations of the labels are accurately adjusted to match the newly sliced dimensions. Given the inherent sparse labeling of the dataset, 96% of the sliced images do not contain the target. Addressing the risk of overfitting caused by a high proportion of negative samples (images without targets), the training and validation datasets are selectively pruned to remove a substantial number of negative samples. This strategy is directed towards creating a more balanced dataset, aiming for an approximate 0.9 : 0.1 ratio between images with and without targets. For the test dataset, selective pruning is omitted to maintain the accuracy and validity of model evaluation.

### 3.5 DATASET RELEASE

The release of the SpaceSet dataset is structured into three distinct subsets, namely SpaceSet-100, SpaceSet-5000, and SpaceSet-full (see Table 1), to cater to varying levels of image processing and analysis requirements.

**SpaceSet-100**: This is the minimal dataset intended for preliminary training and testing purposes. It includes 100 high-resolution images that provide a foundational basis for algorithm development and initial performance assessments. This subset is ideal for quick iteration cycles and for researchers who are beginning their work on SSA without requiring extensive computational resources. SpaceSet-100 is particularly useful for initial model training and validation, performance benchmarking of new methods, and educational purposes, allowing students and new researchers to get hands-on experience with SSA data.

**SpaceSet-5000**: This subset expands the dataset to 5000 images, all captured from Camera 2. It is designed to offer a more comprehensive dataset that can be used for more rigorous training and testing of machine learning models. SpaceSet-5000 provides a larger sample size to improve the robustness of algorithms and to ensure that the models are exposed to a wider variety of scenarios and conditions encountered in space-based observations. It is intended for detailed algorithm development and refinement, robustness testing across a larger set of scenarios, and intermediate-scale projects that require significant but manageable computational resources.

**SpaceSet-full**: This is the full version of the SpaceSet dataset, featuring 5000 images captured from each of the four simulated cameras, resulting in a total of 20,000 images. This comprehensive dataset is collected for advanced research and development objectives. It supports the training and validation of complex models that require multi-view observations to accurately detect and track RSOs. The

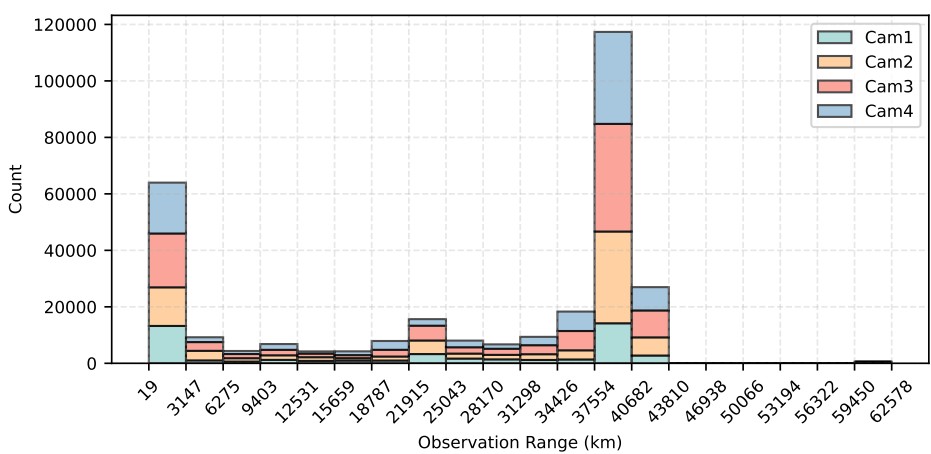

Figure 4: Histogram of observation range for four cameras in SpaceSet-full.

multi-camera setup allows researchers to develop and test algorithms capable of leveraging spatial information from different perspectives, enhancing the accuracy and reliability of SSA systems.

## 4    SpaceSet Dataset Analysis

**Statistical Features of Individual Images:** Here, we provide a specific quantitative description of images of size $4418 \times 4418$ pixels (*16-bit*, *39.1MB* in storage). After testing, the average signal-to-noise ratio (SNR) and average root mean square (RMS) contrast Peli (1990) of SpaceSet-full images are 1.94 dB and 4.67, respectively (typical values for general images are 30 dB for SNR and 80 for contrast). With a threshold pixel value of 2000, the bright point ratio is 0.47. This large image size ensures high resolution and clarity, which is beneficial for detailed analysis and visualization in various applications. However, hardware or applications such as edge computing in satellites may have difficulty handling images with such high resolution. In such cases, we suggest compressing the images before using this dataset. In real-space environments, various types of noise, such as optical and electromagnetic interference, can be present, which manifest in the SNR parameter of this dataset. Generally, a smaller SNR indicates a greater amount of noise in the images (see Figure 3 (b)). This suggests that the image information in this dataset more closely aligns with real-world conditions but poses challenges for feature extraction. Additionally, due to variations in the distance between the targets and the cameras, the brightness of different targets varies, which is a natural phenomenon in imaging. The targets appear brighter when they are closer to the camera and dimmer when they are farther away. As some applications may have specific requirements for image brightness, some image processing algorithms may be needed to enhance brightness.

**Statistical Features of the Whole Dataset:** Figure 4 shows the histogram of observation ranges for four cameras in SpaceSet-full. The horizontal axis of the graph represents the observation distance of the cameras, and the vertical axis represents the counts of target occurrences. Each bar in the histogram corresponds to a range of 3128 km. The closest observed target is at a distance of 19 km, while the farthest is at 62,578 km. The bars for the ranges 19 to 3147 km and 37,554 to 40,682 km are the highest, indicating the highest frequency of target occurrences. In contrast, the frequencies are much lower for other ranges, resembling a long-tail distribution commonly seen in the dataset. More analysis, including the positional distributions and the size distribution of all observed objects, is illustrated in A.1 Figure 5 and Figure 6, respectively.

## 5    Benchmark

As SpaceSet is regarded as a new SSA dataset, we provide several object detection and tracking baseline results based on representative one or two stage detectors and detection-based multiple object tracking methods on SpaceSet-100. All the code for the methods used is available in the sup-

plementary material. Additionally, detailed descriptions of the experimental settings and additional comparison results can be found in Appendix A.

## 5.1 BASIC SETTINGS

We utilize the Ultralytics library Jocher et al. (2023) under the AGPL-3.0 license, which includes all YOLO series models for object detection and tracking (ODT). Other architectures are implemented using the MMDetection toolbox Chen et al. (2019) from OpenMMLab. Following standard configurations, all models are trained on an NVIDIA RTX A6000 GPU for up to 400 epochs, with early stopping if there is no improvement for 30 epochs. From the SpaceSet-100 dataset, 70% of images are used for training and 20% for validation. To minimize randomness in testing, 100 images from the SpaceSet-5000 dataset are selected for evaluation. Metrics for detection performance include mAP@50, mAP@50-95, precision, recall, and F1 score, while object tracking is assessed using false positives, ID switches, and multiple object tracking accuracy (MOTA).

## 5.2 OBJECT DETECTION BENCHMARK

In our object detection benchmark, we evaluated various models, primarily YOLOv8m. We also included yolov3_mobilenetv2, using MobileNetV2 as the YOLOv3 backbone Redmon (2018), and faster_rcnn, based on the ResNet architecture Ren et al. (2016). Transformer-based models such as DETR, Deformable-DETR, and DINO were also employed Carion et al. (2020) Zhu et al. (2020) Zhang et al. (2022a). To optimize training, batch sizes were adjusted based on GPU memory constraints. All models utilized our custom preprocessing pipeline, tailored to the challenges of SSA datasets. All models used 260x260 images as input to ensure result reliability.

The experimental results in Table 2 show the strengths and weaknesses of each method in terms of model size, speed, and accuracy. Transformer-based models capture complex features but require more memory and inference time, making them less suitable for space-based tasks. Faster_rcnn achieves an acceptable level of accuracy but demands significant memory and long detection times, indicating a trade-off between accuracy and efficiency. Yolov3_mobilenetv2 provides faster detection but sacrifices performance, highlighting the challenges of balancing speed and accuracy in SSA.

Existing SOTA methods struggle in the space domain due to the sparse distribution of small targets, many of which occupy only a few pixels in large 4418x4418 images. Over 99% of each image is background noise, presenting challenges not found in typical detection tasks, which emphasizes the unique value of our dataset.

Overall, the YOLOv8m model stands out with its small size, high accuracy, and reasonable detection speed, making it the most suitable for SSA tasks. Its balance of speed and precision effectively meets the unique challenges of space-based monitoring.

Table 2: Performance comparison of SOTA models for space object detection (averaged over three runs). Mem denotes GPU memory usage during training. T/epoch refers to training time per epoch. Epoch indicates the number of epochs until convergence. Size refers to the storage size of trained models. Precision (P), Recall (R), and F1 score are presented as mean ± standard deviation.

| Model Information | Training Process | | | | | Testing Process | | | |
|---|---|---|---|---|---|---|---|---|---|
| Models | batch | Mem | T/epoch | Epochs | Size(MB) | P | R | F1 | T/img |
| yolov3_mobilenetv2 | 48 | 34.36G | 5.09s | 57 | 35.9 | $0.288 \pm 0.005$ | $0.277 \pm 0.023$ | $0.282 \pm 0.010$ | **1.99s** |
| faster_rcnn | 40 | 42.70G | 1.82s | 24 | 333.8 | $0.347 \pm 0.013$ | $0.315 \pm 0.020$ | $0.329 \pm 0.009$ | 6.42s |
| DETR | 8 | 23.18G | 1.78s | 209 | 512.2 | $0.236 \pm 0.055$ | $0.312 \pm 0.026$ | $0.267 \pm 0.043$ | 4.60s |
| deformable_detr | 8 | 38.91G | 4.96s | 141 | 498.8 | $0.315 \pm 0.017$ | $0.479 \pm 0.024$ | $0.380 \pm 0.018$ | 9.71s |
| DINO | 8 | 34.13G | 6.62s | 35 | 597.7 | $0.332 \pm 0.014$ | $0.495 \pm 0.092$ | $0.394 \pm 0.021$ | 13.15s |
| YOLOv8m | 48 | 19.20G | 3.81s | 209 | 52.1 | $0.600 \pm 0.017$ | $0.435 \pm 0.005$ | **$0.492 \pm 0.019$** | 3.81s |

In selecting state-of-the-art (SOTA) models for space object detection, the YOLO series was chosen for its strong performance across various tasks. Given the demands of space deployment, factors such as computational complexity, detection speed, and accuracy are crucial. Various YOLO versions and sizes offer trade-offs between these factors. To identify the optimal model for space deployment, comprehensive experiments were conducted.

As shown in Table 3, three object detection models are compared: YOLOv5 Jocher (2020), YOLOv8 Jocher et al. (2023), and YOLOv10 Wang et al. (2024), each evaluated with five parameter sizes (n, s, m, l, x). Generally, larger models (m, l, x) achieve better accuracy but require more memory and longer training times. While YOLOv8m demonstrated superior performance, smaller models like YOLOv5n and YOLOv8s exhibited lower performance but faster detection times. YOLOv8n balanced speed and accuracy effectively, making it more suitable for SSA, where lightweight computation and timely detection are essential.

Our experiments on the SpaceSet-100 dataset reveal that existing SOTA object detection and tracking methods, effective in conventional scenarios, underperform in space environments, achieving significantly lower scores compared to standard datasets. This underscores the need for algorithms specifically designed for SSA, and our dataset aims to bridge this gap.

Table 3: Detection results of baseline object detection models on SpaceSet-100. During the training of v10x, in order to prevent the GPU memory from being full, we adjusted the batch size to 36.

| Model Information | Training Process | | | | Testing Process | | | |
|---|---|---|---|---|---|---|---|---|
| Models | Mem | T/epoch | Epochs | Size(MB) | P | R | F1 | T/img |
| v5n | **6.38G** | **8s** | 62 | 5.30 | 0.78 | 0.18 | 0.29 | **1.93s** |
| v5s | 10.20G | 26s | 312 | 18.60 | 0.66 | 0.33 | 0.44 | 2.36s |
| v5m | 18.20G | 25s | 226 | 50.50 | 0.72 | 0.26 | 0.38 | 3.45s |
| v5l | 28.00G | 37s | 236 | 106.80 | 0.65 | 0.33 | 0.44 | 4.88s |
| v5x | 40.80G | 63s | 239 | 195.00 | 0.65 | 0.33 | 0.44 | 7.07s |
| v8n | 6.65G | 8s | 236 | 6.30 | 0.65 | 0.36 | **0.47** | 2.01s |
| v8s | 10.60G | 13s | 72 | 22.50 | 0.72 | 0.18 | 0.29 | 2.37s |
| v8m | 19.20G | 25s | 228 | 52.10 | 0.62 | 0.38 | **0.47** | 3.81s |
| v8l | 29.80G | 41s | 305 | 87.70 | 0.55 | 0.38 | 0.45 | 5.55s |
| v8x | 37.40G | 61s | 192 | 136.70 | 0.58 | 0.38 | 0.46 | 7.02s |
| v10n | 9.10G | 11s | 345 | 5.80 | 0.65 | 0.28 | 0.39 | 2.38s |
| v10s | 15.60G | 19s | 85 | 16.50 | 0.67 | 0.27 | 0.38 | 2.63s |
| v10m | 26.30G | 31s | 216 | 33.50 | 0.63 | 0.34 | 0.44 | 3.74s |
| v10l | 40.80G | 48s | 188 | 52.20 | 0.66 | 0.34 | 0.45 | 5.66s |
| v10x | 41.80G | 68s | 242 | 64.10 | 0.65 | 0.36 | 0.46 | 6.54s |

## 5.3 Object Tracking Benchmark

SpaceSet can also be used to evaluate multiple object tracking methods, as it contains ID information for each space object. The ground truth tracking data includes timestamps, object IDs, and two bearing angles. During tracking, pixel positions of the targets are converted into bearing angles using formulas (3) and (4) for comparison with the actual data.

Table 4 summarizes the results of various tracking methods based on a YOLOv8n model with an F1 score of 0.4674. Two tracking methods were tested: Bytetrack Zhang et al. (2022b) and BoT-SORT Aharon et al. (2022), using Intersection over Union (IoU) and Euclidean distance for similarity calculations. Variants of BoT-SORT incorporated different global motion compensation algorithms (ECC, ORB, SIFT, and Sparse Optical Flow), and feature-based similarity calculations were explored using features from the pre-trained YOLOv8n model, as well as traditional methods like HOG and SIFT.

From Table 4, it is evident that the performance of models using IoU distance is far inferior to those using Euclidean distance. This is because space targets are small and fast-moving, and even slight calculation errors can result in an IoU of 0. For fast-moving objects in space, minor camera movements have minimal impact on tracking. Consequently, the evaluation metrics for different global motion compensation methods show little variation in MOTA. BoT-SORT, an improved version of Bytetrack with global motion compensation, performs similarly to Bytetrack because the SSA dataset is insensitive to camera motion. For the SpaceSet dataset, global motion compensation increases processing time with minimal benefits. Although IoU and Euclidean distance calculations

Table 4: Performance evaluation of multiple object tracking methods on SpaceSet-100 (averaged over three runs). Total and Predict columns refer to the total and predicted target numbers, respectively. Real and Predict columns show actual and predicted object numbers. Evaluation metrics include Matches, Misses, False Positives (FP), ID switches (IDs), Multi-Object Tracking Accuracy (MOTA), and tracking time(Time). MOTA are presented as mean ± standard deviation

| Model Information | Target Number | | Object Number | | Evaluation Metrics | | | | | |
|---|---|---|---|---|---|---|---|---|---|---|
| Models | Total | Predict | Real | Predict | Matches | Misses | FP | IDs | MOTA | Time |
| Byte_iou | 4695 | 182 | 56 | 180 | 178 | 4517 | 4 | 143 | 0.0065 ± 0.0002 | 0.06 |
| Byte_euclidean | 4695 | 2111 | 56 | 134 | 2098 | 2597 | 13 | 202 | 0.4012 ± 0.0072 | 0.05 |
| BoT_iou | 4695 | 182 | 56 | 181 | 178 | 4517 | 4 | 143 | 0.0065 ± 0.0002 | 0.05 |
| BoT_euclidean | 4695 | 2107 | 56 | 150 | 2095 | 2600 | 11 | 183 | 0.4049 ± 0.0078 | 0.05 |
| BoT_euclidean_ecc | 4695 | 2101 | 56 | 145 | 2092 | 2603 | 9 | 203 | 0.4002 ± 0.0075 | 1.53 |
| BoT_euclidean_orb | 4695 | 2105 | 56 | 145 | 2095 | 2597 | 9 | 204 | 0.4003 ± 0.0082 | 0.10 |
| BoT_euclidean_sift | 4695 | 2102 | 56 | 139 | 2092 | 2603 | 10 | 196 | 0.4012 ± 0.0085 | 1.15 |
| BoT_euclidean_sparse | 4695 | 2106 | 56 | 145 | 2096 | 2599 | 10 | 202 | 0.4017 ± 0.0089 | 0.14 |
| BoT_feature_yolo | 4695 | 2498 | 56 | 53 | 2486 | 2209 | 12 | 51 | **0.5160 ± 0.0091** | 0.26 |
| BoT_feature_hog | 4695 | 499 | 56 | 57 | 495 | 4200 | 3 | 52 | 0.0938 ± 0.0044 | 7.46 |
| BoT_feature_sift | 4695 | 121 | 56 | 33 | 117 | 4578 | 3 | 4 | 0.0235 ± 0.0010 | 1.42 |

are fast, their accuracy is lower than feature distance based on YOLO. Among all tracking methods, the YOLO feature extraction method performs the best, with a 27% higher accuracy than Euclidean distance. Additionally, traditional feature extraction methods like HOG and SIFT perform poorly.

In addition to the limitations on computing resources and speed, space target tracking tasks must also minimize false detections and ID switches. Excessive false detections and ID switches can negatively impact subsequent tasks such as orbit determination and orbit propagation. BoT-SORT, based on YOLO feature extraction, performs well in all these aspects, making it more suitable for SSA tasks.

## 6 LIMITATIONS OF SPACESET DATASET

Despite the comprehensive nature of the SpaceSet dataset, there are current limitations to consider. First, while the dataset is designed to be highly realistic, the images are still generated via simulations, which may not capture all the complexities and variabilities of real space environments. Second, the dataset focuses on high-resolution images, which, while beneficial for detailed analysis, also require considerable computational resources for processing and storage, potentially limiting accessibility for researchers with limited resources. Lastly, while the dataset includes a range of orbital distances and conditions, it hasn't covered all possible scenarios that SSA systems might encounter, necessitating further validation with real-world data to ensure robustness and generalizability of the developed algorithms.

## 7 CONCLUSION

Focusing on improving SSA, we present SpaceSet, a large-scale realistic space-based image dataset designed to overcome the limitations of existing datasets such as SPARK. SpaceSet provides a comprehensive collection of high-resolution images ($4418 \times 4418$ pixels) generated using accurate space orbit dynamics and a physical camera model with Poisson noise distribution, capturing observations from 19 km to 63,000 km. The dataset is divided into three subsets: SpaceSet-100, SpaceSet-5000, and SpaceSet-full, each catering to different research needs. Our benchmark evaluations show that while larger YOLO models generally outperform smaller ones, lightweight models like YOLOv5s and YOLOv8n offer faster detection speeds, crucial for space-based applications with limited computing resources. Moreover, state-of-the-art object detection and tracking methods perform inadequately in the space environment, underscoring the necessity for algorithms tailored to SSA. SpaceSet not only facilitates the development of new object detection and tracking algorithms but also serves as a benchmark for evaluating advanced SSA techniques.

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

# A APPENDIX

## A.1 MORE ANALYSIS OF RSO DISTRIBUTIONS IN DATASET

Figure 5 shows the positional distribution of targets across four camera datasets. In these maps, the origin of the coordinates is set at the top-left corner of each image, with x and y representing pixel coordinates. The targets are distinguished by their orbital categories: blue points for Low Earth Orbit (LEO) targets, green points for Medium Earth Orbit (MEO) targets, and red points for Geostationary Earth Orbit (GEO) targets.

From the figure, it can be observed that the red and green points form dense, curved trajectories. This is because MEO and GEO targets, being farther from the observer, appear to move more slowly in the images, resulting in closely spaced positions across consecutive frames. In contrast, LEO targets are more widely dispersed, indicating a faster apparent movement across the images.

The figure also shows that there is no discernible pattern in the occurrence and density of the targets. The complex nature of the space environment, combined with variations in observation times, viewing angles, and the orbital paths of tracking instruments, leads to differences in target position distributions. This complexity poses a significant challenge for space object detection.

Since the orientations of the object is neigable considering the far-range observation, the size distribution for all observed objects is illustrated in Figure 6. From all distributions, most of the RSOs are within 0.5m to 10m, and the smallest object is approximately 30cm. Given the large detection range, detecting such small objects in space is extremely challenging, which highlights the significant value of our dataset and benchmark pipeline.

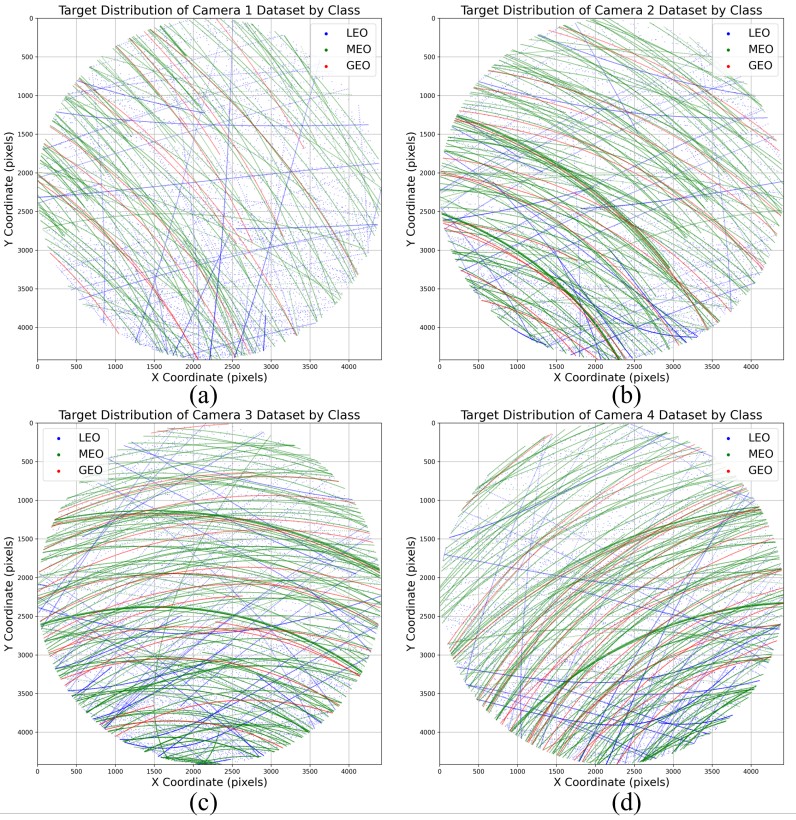

Figure 5: Positional distribution of targets across four camera datasets, categorized by orbital type (LEO, MEO, GEO).

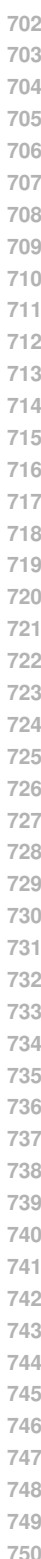
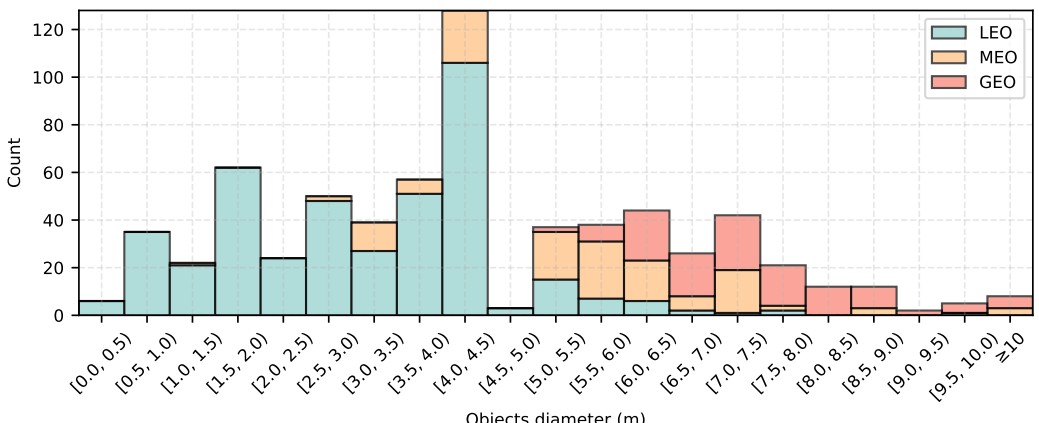

Figure 6: The size distributions for all observed objects categorized by orbital type (LEO, MEO, GEO)

## A.2 COMPARISON WITH EXISTING SPACE DATASETS

Most existing datasets, such as BUAA-SID-share 1.0 Zhang et al. (2010), SPARK Musallam et al. (2021a), and the Space Target Dataset Zhang et al. (2022c), are primarily designed for satellite pose estimation and space target classification. These datasets focus on capturing targets from close distances and multiple viewing angles to highlight specific characteristics. In contrast, SpaceSet captures targets at various distances, simulating realistic space-based camera observations that significantly differ from the approaches of existing datasets.

To emphasize the uniqueness of SpaceSet, we compare its features with those of other existing datasets, as summarized in Table 5. As shown, SpaceSet aims to provide the first large-scale realistic space-based image dataset at the photon level, which advances AI-driven Space Situational Awareness (SSA) techniques. Additionally, the table highlights the size, resolution, and accessibility of each dataset. Currently, fewer than six datasets exist in this domain, all of which are simulated, as NASA's real observations are not publicly available. This lack of accessible real-world data underscores the contribution of SpaceSet to the field.

Table 5: Comparisons of SpaceSet with existing space datasets.

| Dataset | #Images | #Objects | Resolution | Public? |
|---|---|---|---|---|
| BUAA-SID-share 1.0 Zhang et al. (2010) | 9.2k | 20 | $320 \times 240$ | Yes |
| SPARK Musallam et al. (2021a) | 30k | 11 | $1440 \times 1080$ | Request |
| RSONAR Suthakar et al. (2023) | 429 | 3 | $1024 \times 1024$ | No |
| Space Target Dataset Zhang et al. (2022c) | 50k | 46 | Variable | No |
| SpaceSet-100 | 100 | 56 | $4418 \times 4418$ | Yes |
| SpaceSet-5000 | 5k | 414 | $4418 \times 4418$ | Yes |
| SpaceSet-full | 20k | 673 | $4418 \times 4418$ | Yes |

A comparison of sample images from these existing datasets, alongside real ground-based telescope images, is provided in Figure 7 (since real space-based images are currently unavailable). The differences between these datasets, the SpaceSet simulated images, and real-world observations are evident. Furthermore, only BUAA-SID-share 1.0 and SPARK are publicly available, limiting direct comparisons. Due to these limitations and the distinct nature of realistic camera observations, it is likely that training models on these existing publicly available datasets and testing on real images

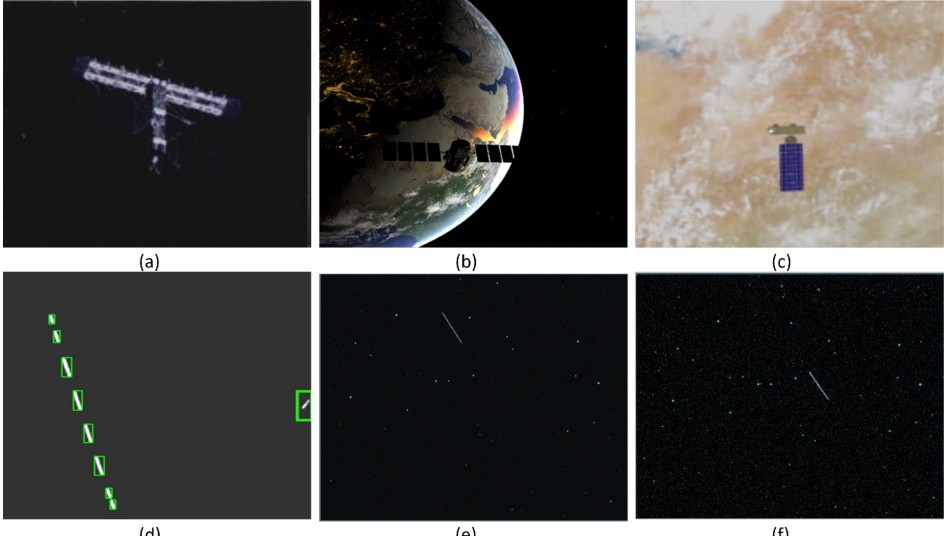

Figure 7: Comparison of images from different datasets. (a) Simulated spacecraft image from BUAA-SID-share 1.0; (b) Simulated spacecraft image from SPARK; (c) Simulated spacecraft image from Space Target Dataset; (d) Simulated space camera observation image from RSONAR, with detected streaks highlighted by green bounding boxes; (e) Simulated space camera observation image from SpaceSet; (f) Real observed images from the telescope at the ground.

would not produce optimal results. As a result, SpaceSet's training performance on other datasets has not been tested, as there are no accessible datasets with comparable space-based observations.

### A.3 IMPLEMENT DETAILS

To ensure the reproducibility of the experimental results, detailed settings for each method used in this paper are provided. The dataset is divided into training, validation, and test datasets with a default partition of 70%, 20%, and 10%. To ensure testing accuracy and minimize randomness, we use 100 new images from SpaceSet-5000, which are the next 100 images in the timestamp sequence following SpaceSet-100, to replace the original test dataset. The original large TIFF image size is $4418 \times 4418$ pixels. The cutting size parameter, which can be adjusted, is set to 260, tiling the large image into smaller $260 \times 260$ pixel images. For the selective pruning process, the adjustable parameter is set to 0.1, creating an approximate 9:1 ratio between images with and without targets. Our models are trained on servers with one Nvidia A6000 GPU (48GB) and an Intel Xeon w9-3495X CPU (4.8 GHz). The open-source codebase YOLOv8 and YOLOv10 can be found at `https://github.com/ultralytics/ultralytics` and `https://github.com/THU-MIG/yolov10`.

### A.4 OBJECT DETECTION METHODS

Using the default settings from Ultralytics Jocher et al. (2023), we train the state-of-the-art object detection models YOLOv5 Jocher (2020), YOLOv8 Jocher et al. (2023), and YOLOv10 Wang et al. (2024). Each model is trained with five different parameter sizes (n, s, m, l, and x). To accelerate the process, all training operations use pre-trained YOLO models released by Ultralytics Jocher et al. (2023). Due to hardware limitations, the SpaceSet-100 dataset is used by default in each experiment. However, the SpaceSet-5000 and SpaceSet-full datasets are also utilized for further experiments, which are detailed in the subsequent More Experiments section. In the training process, some default parameters are adjusted to better fit our scenario, as shown in Table 6. Specifically, the epochs are set to 500 to ensure model convergence. The patience parameter is set to 30, meaning training will automatically stop if there is no improvement for 30 consecutive epochs. The learning rate scheduler is changed from linear decay to exponential decay. Additionally, the number of workers is set to 24 to optimize data loading and accelerate training. For the inference process, the original large image

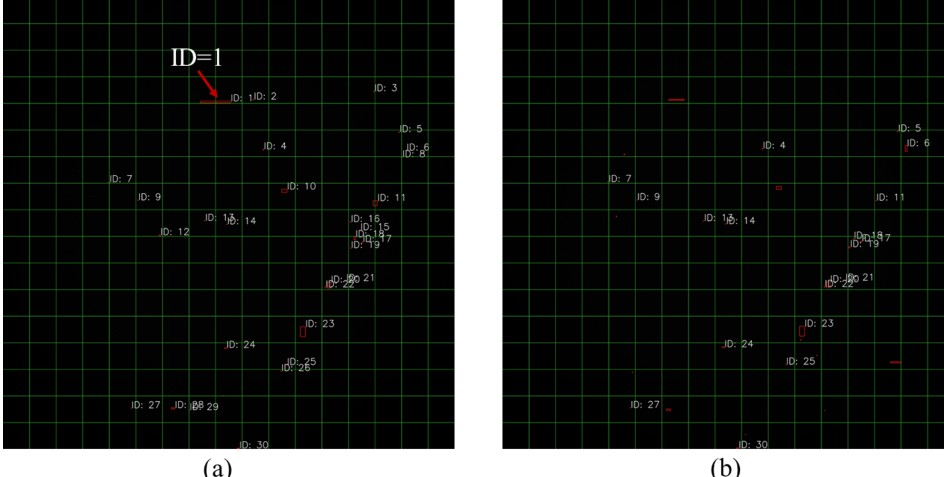

Figure 8: Merging detected objects from segments to the full image for accurate tracking. (a) shows the detection and tracking results at time 0; (b) presents the results at time 1. The original 4418x4418 image is divided into 260x260 segments, as indicated by the green grid, where object detection is initially performed. The detected objects are then mapped back to their positions on the full-size image. Finally, object tracking is performed directly on the full-size image.

is first tiled into smaller images for detection. After detection, these small images and their results are combined back into the original large image. The parameter distance_threshold is used to merge divided targets back into a complete one. Only detection results with an IoU greater than 0.5 are considered valuable.

Table 6: Implement details for object detection benchmark on SpaceSet-100

| Parameter | Value | Description |
| --- | --- | --- |
| epochs | 500 | Number of training cycles to ensure model convergence |
| batch | 48 | Number of samples processed before the model is updated |
| workers | 24 | Number of data loading subprocesses to speed up training |
| lr_scheduler | 'exponential' | Strategy for reducing the learning rate over epoch |
| patience | 30 | Early stopping criteria based on epochs without improvement |
| distance_threshold | 60/4418 | Threshold for merging divided targets in the original image |
| IoU_threshold | 0.5 | Minimum intersection over union for valid detection |

## A.5 OBJECT TRACKING METHODS

All the selected multiple object tracking methods are detection-based, relying on a well-trained YOLOv8n model, as described in Section 5.2 Object Detection Benchmark. The tracking process operates on the original full-size images with a resolution of 4418x4418 pixels. To facilitate detection, each image is initially divided into 260x260 segments, and the position of each segment within the original image is recorded. Object detection is then performed on these segments, and the detected positions are mapped back to their respective locations on the original full-size image, using the recorded positional information.

To handle objects that may span across adjacent segments, a merging strategy is implemented. The strategy calculates the distance between the edges of bounding boxes in neighboring segments. If the distance between the right edge of one bounding box and the left edge of an adjacent one is below a set threshold, the bounding boxes are merged, ensuring an accurate representation of the object on the original image. After this merging process, the tracking algorithm operates on the entire original

image, avoiding issues such as objects moving out of segment boundaries and enhancing tracking accuracy.

The tracking process incorporates the Euclidean distance calculation and feature distance calculations using three feature extraction techniques to improve robustness. The experimental settings for tracking largely follow those in Ultralytics Jocher et al. (2023), with specific differences listed in Table 7. The parameter tracker_type can be set to either 'botsort' or 'bytetrack', representing two tracking algorithms, and gmc_method is set to 'none' to ensure faster tracking, as GMC has minimal impact on the SSA dataset.

Table 7: Implement details for object tracking benchmark on SpaceSet-100

| Parameter | Value | Description |
|---|---|---|
| tracker_type | 'botsort' | Type of tracking algorithm ('botsort' or 'bytetrack') |
| gmc_method | 'none' | Options include 'orb', 'sift', 'ecc', 'sparseOptFlow', 'none' |
| with_reid | True | Use feature distance (True or False) |
| feature_method | 'yolo' | Method for feature extraction ('yolo', 'hog', 'sift') |
| angle_threshold | 0.2 | Threshold for matching tracking results with ground truth |
| track_low_thresh | 0.4 | Threshold for considering low-confidence detections |

The with_reid parameter, set to True, indicates the use of feature distance, which can be computed using 'yolo', 'hog', or 'sift' as the feature_method. Additionally, the angle_threshold is set to 0.2, to match tracking results with ground truth, considering the ID correct only if the difference in bearing angles is below this threshold. The track_low_threshold is set to 0.4, allowing the tracking algorithms to consider low-confidence detections to reduce misses and ID switches.

The detection and tracking process across consecutive frames is illustrated in Figure 8 of the supplementary material. Detection is performed within the 260x260 segments (grid division shown in green), and objects spanning segments are merged. Once detection and merging are complete, tracking is performed on the full original image to ensure reliable tracking performance across the entire field of view.

## A.6 Comparison with Real-World Ground-Based Telescope Images

To evaluate the model trained on the simulated dataset, tests were conducted using real-world images obtained from a ground-based telescope. The ground-based telescope used is a Celestron CPC RASA with a 620mm focal length, located at coordinates 32.9024°N, -105.53202°W, equipped with a QHY268M camera (3.76 micron pixel size, 26 Megapixel resolution). Due to the current unavailability of real space-based observational data, these ground-based images serve as the best available alternative for assessing model performance on real data. Figure 9 illustrates the detection results in a real-world image.

The model successfully detected the most prominent linear object in the real-world image. Several other objects were also identified, though some detections may be false positives. These false detections are likely caused by differences between ground-based telescope images and space-based camera observations, particularly due to atmospheric conditions that affect the spot size and distribution in the images. Such discrepancies highlight the challenges of directly transferring models trained on simulated space-based data to ground-based observations.

To provide a more detailed comparison, Table 8 compares the Right Ascension (RA) and Declination (Dec) values between real observations and our simulated images, showing the small differences in pointing direction and starfield representation. The comparison also accounts for additional effects from the atmosphere in ground-based images, especially differences in spot size.

Furthermore, Table 9 compares the tracked positions of objects from the ground-based telescope and the SGP4 propagation simulator. With the given pointing direction, the object positions at different timestamps are nearly identical to the real-world observations. The results demonstrate the accuracy of our simulator regarding the observation pointing direction and object positions.

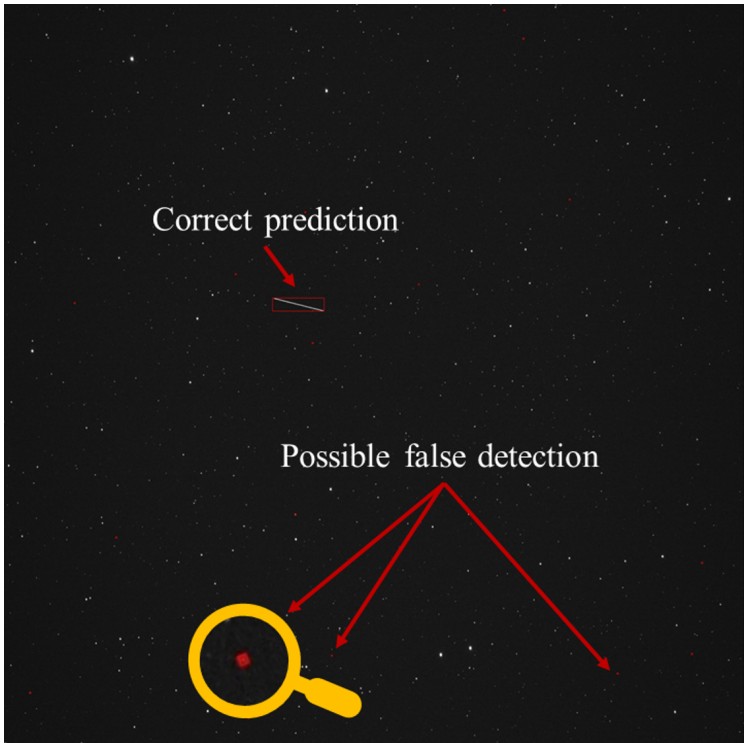

Figure 9: Detection results on a real-world ground-based telescope image using a model trained on simulated space-based data with Zoomed-in view of a potential false positive.

Table 8: Comparison of RA and Dec values from different sources for starfield.

| Source | Telescope | `http://astrometry.net/` | Simulator | Difference |
|---|---|---|---|---|
| RA (deg) | 334.900806 | 334.888 | 334.911317 | 0.011 |
| Dec (deg) | 57.236457 | 57.223 | 57.2352077 | 0.00125 |

The comparison of generated images is illustrated in Figures 10 and 11. The observed similarities confirm that the simulated images effectively replicate key characteristics of real space-based observations, such as streaks of fast-moving RSOs and noise distributions, despite the inherent differences caused by atmospheric interference in ground-based telescope images.

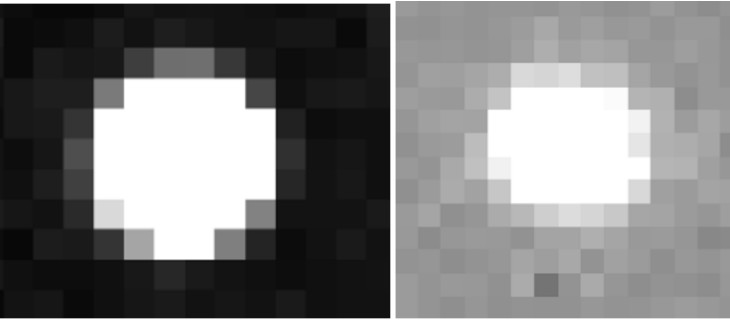

Figure 10: The spot comparison of our simulated image (left) and the real observed image with a ground-based telescope (right). The star has a visual magnitude of 8.6.

Table 9: Comparison of tracked position from the telescope and SGP4 propagation simulator.

| | Telescope | Simulator | Difference |
|---|---|---|---|
| **Time Stamp** | 2023-12-25T01:40:20.97 | 2023-12-25 01:40:20.999994 | |
| RA (deg) | 334.724 | 334.765 | 0.041 |
| Dec (deg) | 57.458 | 57.380 | 0.078 |
| **Time Stamp** | 2023-12-25T01:40:22.515 | 2023-12-25 01:40:22.513506 | |
| RA (deg) | 335.003 | 335.037 | 0.034 |
| Dec (deg) | 56.580 | 56.520 | 0.060 |

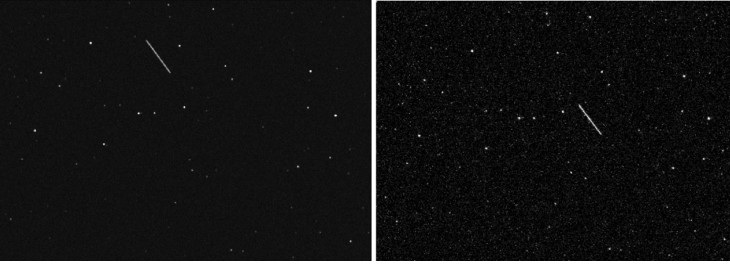

Figure 11: The image comparison of our simulated image (left) and the real observed image with a ground-based telescope (right). These simulation parameters are matched to the real observation position and pointing directions.

In practice, false detections are typically filtered out during the tracking process. Objects detected in only a single frame, which cannot be matched across multiple frames, are classified as false detections and removed. However, the lack of consecutive real images from ground-based observations limited the ability to test tracking performance comprehensively on real-world data.

This evaluation highlights the correctness of our simulated image dataset SpaceSet, which aims to bridge the gap and provide realistic space-based observational data for model training and testing.

## A.7 MORE EXPERIMENTS

To demonstrate the performance of object detection and tracking methods on the SpaceSet-5000 and SpaceSet-full datasets, additional experiments are conducted. Due to hardware limitations, only one object detection method (YOLOv8m) and one multiple object tracking method (BoT_feature_yolo) are used in these experiments. For the SpaceSet-5000 dataset, the detection results are presented in Table 10, and the tracking results are shown in Table 11. The parameter GT represents the total number of targets in the ground truth, while Total Predict indicates the total number of targets predicted by the model. Correct Predict is the total number of correctly predicted targets. The results in the table demonstrate that the YOLOv8m model achieves high detection accuracy with a sufficiently large dataset. The training process follows a 0.7:0.2:0.1 split for the training, validation, and test datasets. The evaluation metrics for the multiple object tracking algorithm show the effectiveness of the feature distance calculation method, based on YOLO feature extraction, in tracking RSOs.

Table 10: Detection results of the object detection model on SpaceSet-5000.

| Model | GT | Total Predict | Correct Predict | Precision | Recall | F1 Score |
|---|---|---|---|---|---|---|
| YOLOv8m | 75376 | 71565 | 56184 | 0.78 | 0.75 | **0.76** |

Similarly, for the SpaceSet-full dataset, the detection results are provided in Table 12, and the tracking results are listed in Table 13. Due to varying pointing directions between different cameras, the captured images exhibit a degree of distribution shift. Additionally, training on multiple camera

Table 11: Tracking results of the object tracking model on SpaceSet-5000.

| Model | Target Number | | Object Number | | Evaluation Metrics | | | | |
|---|---|---|---|---|---|---|---|---|---|
| Name | Total | Predict | Real | Predict | Matches | Misses | FP | IDs | MOTA |
| BoT_feature_yolo | 75376 | 55350 | 414 | 431 | 54965 | 20816 | 385 | 406 | **0.71** |

Table 12: Detection results of object detection model on SpaceSet-full.

| Dataset | GT | Total Predict | Correct Predict | Precision | Recall | F1 Score |
|---|---|---|---|---|---|---|
| Cam1 | 44683 | 51631 | 31203 | 0.61 | 0.70 | 0.65 |
| Cam2 | 75376 | 76260 | 52839 | 0.69 | 0.70 | **0.70** |
| Cam3 | 97204 | 85684 | 62542 | 0.72 | 0.64 | 0.68 |
| Cam4 | 84614 | 81314 | 58256 | 0.71 | 0.69 | **0.70** |

datasets may degrade the model's performance on a single camera dataset. This issue introduces the need to explore ways to enhance model generalization and continual learning capabilities in the SSA field. Such exploration can promote the development of space object detection and tracking algorithms with higher generalization capabilities.

Table 13: Tracking results of the object tracking model on SpaceSet-full.

| Dataset | Target Number | | Object Number | | Evaluation Metrics | | | | |
|---|---|---|---|---|---|---|---|---|---|
| Name | Total | Predict | Real | Predict | Matches | Misses | FP | IDs | MOTA |
| Cam1 | 44683 | 26933 | 373 | 376 | 26464 | 18612 | 469 | 219 | 0.57 |
| Cam2 | 75376 | 51770 | 414 | 404 | 51503 | 24278 | 267 | 357 | **0.67** |
| Cam3 | 97204 | 60952 | 452 | 498 | 60792 | 36857 | 160 | 584 | 0.61 |
| Cam4 | 84614 | 56433 | 426 | 623 | 55915 | 39119 | 518 | 616 | 0.64 |

