# OpenReview forum: "SpaceSet: A Large-scale Realistic Space-based Image Dataset for Space Situational Awareness"
_ICLR.cc/2025/Conference — Submitted to ICLR 2025_

### Official Review · Reviewer_ajKv · 2024-10-27

**Soundness:** 4
**Presentation:** 4
**Contribution:** 4
**Rating:** 10
**Confidence:** 1

**Summary:**

Given my limited expertise in the field of space observations, I find myself ill-equipped to provide a comprehensive evaluation of this paper. The specific challenges and nuances within this domain are not within my area of specialization. Therefore, I recommend that you consult with a diverse group of reviewers who possess a deeper understanding of space-related research to ensure a thorough and informed assessment of the paper's content and significance.

**Strengths:**

Given my limited expertise in the field of space observations, I find myself ill-equipped to provide a comprehensive evaluation of this paper. The specific challenges and nuances within this domain are not within my area of specialization. Therefore, I recommend that you consult with a diverse group of reviewers who possess a deeper understanding of space-related research to ensure a thorough and informed assessment of the paper's content and significance.

**Weaknesses:**

Given my limited expertise in the field of space observations, I find myself ill-equipped to provide a comprehensive evaluation of this paper. The specific challenges and nuances within this domain are not within my area of specialization. Therefore, I recommend that you consult with a diverse group of reviewers who possess a deeper understanding of space-related research to ensure a thorough and informed assessment of the paper's content and significance.

====

**Post-Rebuttal**: I think it is a good topic in AI4Science. However, my expertise does not extend into the realm of space observations. Consequently, I am not in a position to accurately assess the quality of the data or the principal challenges it presents. Therefore, I must express my recommendation for this paper with a degree of caution and limited confidence, acknowledging the limitations of my own knowledge in this specific area.

**Questions:**

Given my limited expertise in the field of space observations, I find myself ill-equipped to provide a comprehensive evaluation of this paper. The specific challenges and nuances within this domain are not within my area of specialization. Therefore, I recommend that you consult with a diverse group of reviewers who possess a deeper understanding of space-related research to ensure a thorough and informed assessment of the paper's content and significance.

---

> ### Author Response · Authors · 2024-11-27
> **Response to Reviewer ajKv**
>
> We sincerely thank your kind feedback on our work. We would like to introduce our work more simply for understanding. We contribute a dataset of space objects in low, middle, and geostationary orbits (observed from 19km to 63,000km in space) in order to enhance research in space situational awareness (SSA) which monitors the motion of RSOs. The full dataset consists of 5,000 images in 4 camera views (20,000 images total), with images simulating noise distributions and camera representative of real observations. Experiments are conducted on object detection and object tracking. We make a significant dataset contribution to the community to address the concern of the lack of datasets representative of real-world space camera observations (currently, there are fewer than six datasets in this field, all simulations, as NASA has not made its database public), and bridging the gap from simulated to real-world data. We hope our dataset and benchmark can attract more attention and interest from the LR and AI community to push the advancement of computer vision and SSA technologies.

---

> > ### Comment · Reviewer_ajKv · 2024-11-28
> >
> > Thank you for your explanation. I am interested in understanding the distinctions between natural images and those captured through space observation. Additionally, I would like to inquire about the specific cosmic rays that could potentially impact space-based observations. Could you elaborate on the types of cosmic rays and the specific effects they have on space observation equipment?
> >
> > I am particularly curious about the following aspects:
> >
> >  - The characteristics that differentiate natural images from those obtained through space observation.
> > - The types of cosmic rays that are known to interfere with space-based imaging.
> > - A detailed account of the efforts being made to mitigate the impact of these cosmic rays on space observation.
> >
> > Your expertise in this area would be invaluable in shedding light on these matters.

---

> > > ### Comment · Reviewer_ajKv · 2024-11-28
> > >
> > > What do you identify as the primary challenges when comparing data from simulated environments to real-world data, especially in the context of space observation? In the process of transferring knowledge or models from simulated environments to real-world applications (Sim2Real), what is considered the most significant challenge?

---

> > > > ### Author Response · Authors · 2024-11-28
> > > > **Response to Reviewer ajKv**
> > > >
> > > > >**Q4** The primary challenges in comparison
> > > >
> > > > The primary challenge we faced when comparing our data to real-world data is the difficulty in obtaining real-world observations. To obtain real-world data, we planned the observation site and pointing directions in simulation and sought our collaborator to acquire observed images from a ground-based telescope with the same configurations for comparison, as described in A.6. To compare our data with space observations, we searched the literature and compared with the images in [5], but with different observation configurations. It took great effort to validate our dataset. Even so, we cannot capture all the complexities and variabilities of real space environments, as we discussed in our limitations. Space observation requires much more effort from governments, space institutes, and companies. In the near future, we aim to collect more real-world space observation data from our collaborators.
> > > >
> > > > >**Q5** The most significant challenge in Sim2Real
> > > >
> > > > In transferring knowledge or models from simulated environments to real-world applications, the most significant challenge is ensuring that models generalize effectively despite data biases as we found from our test in Figure 9. This involves addressing domain shifts, where the statistical properties of simulated data differ from those of real data, even though our dataset is the most realistic one. Overcoming this challenge requires using domain adaptation techniques or continual learning, and fine-tuning models with small real-world dataset to bridge the gap and enhance performance in practical applications.

---

> > > ### Author Response · Authors · 2024-11-28
> > > **Response to Reviewer ajKv**
> > >
> > > Thank you for your interests and insightful questions. We are pleased to provide detailed explanations to address your concerns.
> > >
> > > >**Q1** Differences between natural images and space observation images
> > >
> > > Natural images are photographs captured on Earth, influenced by atmospheric conditions, diverse lighting, and complex backgrounds. They often contain varied textures and colors, with noise primarily arising from sensor limitations such as thermal noise, read noise, and shot noise due to low light. In contrast, space observation images are taken outside Earth's atmosphere, resulting in high contrast between celestial objects and the dark background of space. The absence of atmospheric scattering leads to sharper images of stars and planets. RSOs are captured by the reflection of starlight, resulting in weak contrast and appearing as streaks due to their high-speed motion, as seen in our dataset. However, space images are subject to unique noise distributions, including cosmic ray hits that appear as random bright pixels or streaks caused by high-energy particles impacting the sensor. Other noise sources in space imaging include dark current noise from thermal electrons in the sensor and radiation-induced defects. These factors lead to increased background noise and hot pixels over time, which we have modeled in our work based on data from the Hubble Space Telescope and the NEOSSAT mission.
> > >
> > > >**Q2** Types of cosmic rays affecting space-based imaging
> > >
> > > The main cosmic rays interfering with space-based imaging are Galactic Cosmic Rays (GCRs) and Solar Energetic Particles (SEPs). GCRs originate outside the solar system and consist of high-energy protons and heavy nuclei. SEPs are emitted by the Sun during solar flares and coronal mass ejections, comprising high-energy protons and electrons. When these particles strike imaging sensors, they can cause transient bright spots or streaks in images (cosmic ray hits) and contribute to long-term sensor degradation due to cumulative radiation damage.
> > >
> > > >**Q3**  Mitigation efforts for cosmic ray impact on space observation
> > >
> > > Mitigating cosmic ray effects involves hardware, software, and operational strategies. Hardware solutions include using radiation-hardened electronics and shielding sensitive components with materials like aluminum to absorb or deflect high-energy particles. Sensor designs may feature redundancy and error-correcting codes to handle radiation-induced errors. Software techniques employ image processing algorithms to detect and remove cosmic ray artifacts, such as filtering out obvious anomalous pixels or stacking multiple images to distinguish real objects from noise. Operational strategies involve planning observation windows during periods of low solar activity and selecting orbits like Sun-Synchronous Orbit in our work and suitable pointing directions that minimize exposure to high-radiation zones.

---

### Official Review · Reviewer_qNXp · 2024-11-02

**Soundness:** 3
**Presentation:** 3
**Contribution:** 2
**Rating:** 5
**Confidence:** 3

**Summary:**

This paper introduces SpaceSet, a large-scale, realistic image dataset designed to enhance space situational awareness (SSA) for tracking and monitoring resident space objects (RSOs). Unlike previous datasets, SpaceSet incorporates accurate orbital dynamics and a physical camera model with photon-level noise distributions to produce realistic space-based images. Simulated from multiple orbital perspectives (LEO, MEO, GEO), the dataset covers distances from 19 km to 63,000 km and provides high-resolution images (4418 × 4418 pixels) suitable for advanced SSA methods. It includes three subsets—SpaceSet-100, SpaceSet-5000, and SpaceSet-full—addressing various image processing needs, along with a benchmark evaluation for detection and tracking algorithms.

**Strengths:**

The main contribution of this paper is a large-scale, realistic image dataset designed to enhance space situational awareness (SSA) for tracking and monitoring resident space objects (RSOs). The dataset has a few merits compared to existing ones:

+ The proposed dataset incorporates realistic orbital dynamics and a camera model with photon-level noise, enhancing its applicability to real-world SSA tasks.

+ The image resolution in the proposed dataset is high (4418 × 4418 pixels). It covers multiple orbital altitudes (LEO, MEO, GEO) and observation distances (19km to 63,000km)

**Weaknesses:**

My major concern is the limited contribution. A pure dataset contribution may not align with the topic of the ICLR conference (Learning Representations). This paper benchmarks many existing detection and tracking algorithms on the proposed subset (SpaceSet-100). However, no new algorithms regarding learning representations are provided.

Another concern is the dataset setup. I understand the proposed one is already closer to real settings than previous ones, but it is still a synthetic dataset. I wonder what the gap is between the proposed synthetic and realistic settings. For example, the dataset uses a fixed camera setup, four overlapping cameras, and a fixed rotation angle. Is this always the real application scenario? Will any missions require a different number of cameras with other relative pose settings (e.g., unstructured)?

Is there any available real dataset that could be used to evaluate the quality of the synthetic data or the synthetic-to-real generalization ability of an algorithm?

Some illustrations might be clearer. For example, from Fig.1, I cannot figure out why (a) shows the realistic exposure with noise distribution, how exposure is reflected, what the noise distribution is, and why they are realistic. (b) shows a picture from an existing dataset, SPARK, but there isn't an image of the proposed dataset for comparison. (c) is almost black and I do not know where to put my focus.

**Questions:**

Plz refer to my comments in "Weakness".

---

> ### Author Response · Authors · 2024-11-27
> **Response to Reviewer qNXp**
>
> We sincerely thank you for the thorough review and constructive feedback. We would like to appreciate your praise on "the dataset has a few merits compared to existing ones". We hope our response can address all your concerns.
>
> >**Q1** Contribution of work
>
> We sincerely understand your concern. However, we submitted our paper to the Primary Area of Datasets and Benchmarks. Currently, there are fewer than six datasets in this field, all simulations, as NASA has not made its database public. We contributed the first large-scale realistic space-based image dataset from the photon level and verified it with ground observations, thereby bridging the gap from simulated to real-world data and addressing an intriguing topic in aerospace and low Earth orbit monitoring. We have made a significant dataset and benchmark contribution to the community to address the concern of the lack of datasets representative of real-world space camera observations. This dataset and our proposed benchmark pipeline provide valuable understanding of existing detection and tracking algorithms and hence push the advancement of Learning Representations of image observation in space, considering the sparse information of the images and various noises.
>
> >**Q2** Dataset setup
>
> To have a good imaging window, the satellite should be operating in a Sun-Synchronous Orbit, and cameras need to maintain an unobstructed line-of-sight (LOS) that avoids the Sun, Earth, and Moon to prevent a high-brightness background. The rotation angle is very limited to provide a good observation window, and our four overlapping settings aim to increase the observation window for detecting objects. We can have other unstructured settings or even utilize a satellite swarm to conduct collaborative observations, but this brings more complexity to the process of object tracking across different images and timestamps.
>
> >**Q3** Data validation and synthetic-to-real generalization
>
> Currently, there are very few real-world in-space images released, and no public real-world in-space datasets available on the market. We have compared our simulated images with the real observed images from ground-based telescopes in Appendix A.6. These results clearly validate the realistic characteristics of our simulated images. The small discrepancies arise from atmospheric effects. Some real in-space observation images from a camera not optimized are released in [5], which discusses the detected images and issues such as  readout at the edge of the sensor and the fringing, and possible solutions to optimize their sensors. From their images, we can also clearly see the similarity regarding the streaks and noises to our dataset and the potential to use our dataset to develop object detection and tracking algorithms to advance SSA technologies. We also conducted tests using a model trained on our simulated dataset, with results on a real-world image illustrated in Figure 9. From the detection results in Figure 9, it is evident that the most prominent streak objects were successfully detected. Additionally, the model detected several other objects, some of which may be false positives. These false detections likely arise from differences in characteristics and distribution between ground-based telescope images and space-based camera images affected by atmospheric conditions.
>
> >**Q4** Clearer illustrations and explanation
>
> Figure 1 aims to show our four camera images (a), the existing simulated spacecraft image from SPARK (b), and the real-world observed image from an in-space telescope (c). Considering the page size, a more detailed comparison is illustrated in Figures 3 and 7 in the revision. Comparing our image (a) and the real observed image (c), the most important features are the streaks (lines) due to the exposure of fast-moving RSOs and the hot pixels induced by the noises. The noise distributions are detailed in Section 3.1 Data Generation (lines 192–200), including background noise, camera shot noise, sensor noise, and read noise. However, the synthetic image from SPARK (b) is far from the real-world observations (c), making it less useful for advancing learning representations for SSA. We have revised the description in Figure 1 to enhance understanding of the figures.

---

> ### Author Response · Authors · 2024-11-30
>
> Dear Reviewer qNXp,
>
> Thank you once again for taking the time to review our work. We greatly appreciate your feedback and have carefully considered your comments. If you have any remaining concerns after reviewing our responses, we would be happy to discuss them further. Please let us know if you have received our responses and if we have successfully addressed your concerns, as tomorrow is the deadline. Thank you again for your valuable suggestions. Have a great day!

---

> ### Author Response · Authors · 2024-12-02
>
> Dear Reviewer qNXp,
>
> Thank you once again for taking the time to review our work. We greatly appreciate your feedback and have carefully addressed your comments. If you have any remaining concerns after reviewing our responses, we would be happy to discuss them further. Please let us know if you have received our responses and if we have successfully addressed your concerns. Additionally, we would appreciate any insights regarding a potential raise in the rating score, as tomorrow is the deadline.
>
> Thank you again for your valuable suggestions, and wish you a great day!

---

> > ### Comment · Reviewer_qNXp · 2024-12-03
> >
> > Thanks to the authors for the response. I don't have other questions.

---

> > > ### Author Response · Authors · 2024-12-03
> > >
> > > Thank you for your time and valuable feedback. We would appreciate it if you could raise the rating score if we have addressed all your concerns. Have a great day!

---

### Official Review · Reviewer_EGtv · 2024-11-04

**Soundness:** 2
**Presentation:** 2
**Contribution:** 2
**Rating:** 5
**Confidence:** 4

**Summary:**

To improve Space Situational Awareness (SSA), the authors introduced SpaceSet, a comprehensive, large-scale dataset of high-resolution spaced-based images, designed to address the limitations of existing simulated datasets. This datasets consists of images generated with accurate orbital dynamics and a physical camera model with various noise distributions, capturing observations from altitudes between 19 km and 63,000 km. In experimental section, the authors provided the object detection and tracking benchmarks. The benchmark indicates that a series of YOLO-8 can show the better performance in the computational overhead and the accuracy.

**Strengths:**

First of all, the motivation of the paper seems to be meaningful and pragmatic in the perspective of addressing limitations in the existing dataset.

SpaceSet contains high-resolution scenes and more RSOs than existing datasets and also contains multi-camera images that consider various physical factors including noise, object properties, camera models, and locations.

To validate its effectiveness as a benchmark for object detection and tracking, the authors selected the most widely used models and conducted meaningful experiments.

**Weaknesses:**

As a benchmark paper, it lacks many details and analysis on dataset itself. Except for data generation, the information such as the properties of RSOs such as size and orientations, the positional distribution of frequently appearing locations in the images, and criterion for image splits is needed. Furthermore, there is a lack of detail regarding the benchmark experiments such as whether all models used input images of the same size or why do existing SOTA object detection and tracking methods not perform as well as expected on SpaceSet-100?

Another concern is the completeness of the paper. Although the completeness is an essential aspect of a paper, this paper contains awkward sentences, typos (e.g., L.140, 210, obit), grammatical errors (e.g., L.216 box are -> is). Regardless of whether these issues hinder understanding of the content, the lack of attention to such things is problematic.

Lastly, the content in A.5 intended to validate the proposed dataset’s effectiveness in real-world data is not clearly conveyed. It would be beneficial to emphasize in the main paper why these metrics provided in Table.8 and 9 are meaningful.

**Questions:**

I mentioned all comments including reasons and suggestions in the above sections. I recommend that the author will provide all the concerns, and improve the completeness of the paper. If the rebuttal period resolves the above-mentioned concerns, I will gladly raise my score. Also, there are many vague sentences and grammatical errors in the paper. I recommend that the author will revise the paper.

---

> ### Author Response · Authors · 2024-11-27
> **Response to Reviewer EGtv**
>
> Thank you for your great efforts in reviewing our submission and providing very helpful suggestions to improve our work. We are happy to receive your praise comments on our wok "being meaningful and pragmatic in the perspective of addressing limitations in the existing dataset".
>
> >**Q1** More details and analysis on dataset.
>
> We have conducted more analysis on the dataset beyond the Signal Noise Ratio (SNR) and the observation range. Since the orientations of the object are negligible considering the far range observation, we have included the size distributions for all observed objects by classes. From all distributions, most of the RSOs are within 0.5 m to 10 m, and the smallest object is approximately 10 cm. Given the large detection range, detecting such small objects in space is extremely challenging, which highlights the significant contributions of our dataset and our benchmark pipeline.
>
> We also plotted positional distribution of all targets in the four camera datasets in Figure 5. The targets are distinguished by their orbital categories: targets in LEO are represented by blue points, targets in MEO by green points, and targets in GEO by red points. From the maps, it can be observed that the occurrence and density of targets do not follow any discernible pattern. Due to the highly complex nature of the space environment, variations in observation times, angles, and the orbital paths of the trackers lead to differences in the distribution of target positions. This is one of the challenges in space object detection.
>
> Regarding the criterion for image splits, we initially experimented with various image sizes, including 260x260, 640x640, and 1280x1280. As the image size increased, the computational load and memory requirements rose significantly. Since most of our targets are small, larger image sizes were not conducive to their detection. With larger images, the model's detection accuracy declined sharply, and beyond a certain point, the model was unable to learn any meaningful information. Meanwhile, smaller image sizes had a negative impact on detection speed, and further reducing the size risked splitting more targets across multiple images, which affected the training. Based on all experimental results, we chose 260x260 as the final split image size, as it provided a good balance between detection accuracy and speed.
>
> >**Q2** More detail regarding the benchmark experiments.
>
> We have included more details of our experiments and revised our paper accordingly. To ensure consistency and reliability in our comparisons, all experiments followed the same settings, including the input images. All models used 260x260 images, which were processed with the same data preprocessing steps, as input.
>
> Existing SOTA  algorithms were originally developed for typical detection tasks in real-world settings. When applied to sparse, small-object detection in the space domain, it is understandable that the performance may fall short of expectations. Unlike conventional small-object detection, some of our targets occupy only a few pixels, and our images are quite large at 4418x4418 resolution. This results in an extremely sparse distribution of targets, with more than 99% of each image being background noise. These challenges are not present in traditional object detection datasets, which highlights the importance of our work. Our dataset provides valuable image data for AI applications in space and introduces new challenges to the computer vision community.
>
> >**Q3**  Proofread of paper
>
> We have thoroughly proofread our paper in the revision to avoid any typos, grammatical errors, and unclear descriptions. We hope our paper now meets the high standards of quality.
>
> >**Q4** Clarification of A.5 and Tables in A.5
>
> Table 8 and Table 9 are used to validate the high accuracy of the orbit propagation simulator, which can generate objects in the correct positions. Table 8 showcases that the simulator provides the exact same starfield (a region of the sky containing stars as seen through a telescope or recorded on an image) for the given station position, time, and pointing direction as the real-world telescope observations. Table 9 validates that, with the given pointing direction, the objects' positions at different timestamps are nearly identical to the real-world observations, where Right Ascension (RA) and Declination (Dec) of an object specify its position uniquely on the celestial sphere. These two tables demonstrate the accuracy of our simulator regarding the observation pointing direction and object positions. With this accurate information, we further compared the simulated images and the real-world observed images from the telescope in Figures 10 and 11 (in the revision). The streaks due to the exposure of high-speed motion and the noise distributions show the effectiveness of the generated dataset. We have added and emphasized this information in the main paper to enhance the understanding of our comparison.

---

> ### Author Response · Authors · 2024-11-30
>
> Dear Reviewer EGtv,
>
> Thank you once again for taking the time to review our work. We greatly appreciate your feedback and have carefully considered your comments. If you have any remaining concerns after reviewing our responses, we would be happy to discuss them further. Please let us know if you have received our responses and if we have successfully addressed your concerns, as tomorrow is the deadline. Thank you again for your valuable suggestions. Have a great day!

---

> ### Author Response · Authors · 2024-12-02
>
> Dear Reviewer EGtv,
>
> Thank you once again for taking the time to review our work. We greatly appreciate your feedback and have carefully addressed your comments. If you have any remaining concerns after reviewing our responses, we would be happy to discuss them further. Please let us know if you have received our responses and if we have successfully addressed your concerns. Additionally, we would appreciate any insights regarding a potential raise in the rating score, as tomorrow is the deadline.
>
> Thank you again for your valuable suggestions, and wish you a great day!

---

### Official Review · Reviewer_ngxj · 2024-11-05

**Soundness:** 3
**Presentation:** 3
**Contribution:** 3
**Rating:** 6
**Confidence:** 2

**Summary:**

This paper proposes a large-scale realistic space-based image dataset for space situational awareness. The dataset contains 20k images with 673 objects. The paper describes the detailed pipeline of data curation and data annotation. Additionally, the authors construct an object detection and tracking benchmark based on the proposed dataset and provide several baseline results.

**Strengths:**

1. This paper proposes a new large scale dataset for space situational awareness. Comparing to previous public avaiable dataset available in this filed, the proposed dataset is the first realistic image dataset at the photon level.
2. This paper constructs object detection and tracking benchmark on the proposed dataset, and implemented several baselines methods on the dataset.

**Weaknesses:**

1. Considering the relatively small number of target objects (673) in the dataset, it is crucial for the authors to perform multiple experimental runs for object tracking tasks to obtain statistical mean values and variances. The authors should ensure that the variation in evaluation metrics across different runs (e.g. 3 or 5 runs) remains below a specified threshold (e.g., 0.5) to establish the reliability and stability of the results.
2. Given that object scale is a critical factor influencing detection performance, the authors should provide comprehensive statistical analysis of object scales in the dataset, such as the distribution of object sizes (e.g., small, medium, and large objects). This information would help readers better understand the dataset characteristics and evaluation results.

**Questions:**

1. Please provide multi-run results with mean and variances on the object detection and object tracking benchmarks to ensure the variances of multiple-runs are small.
2. The authors could include baseline results of specialized small object detection methods, such as approaches [1] that have shown promising performance on the VisDrone [2] benchmark.

[1] QueryDet: Cascaded Sparse Query for Accelerating High-Resolution Small Object Detection.

[2] VisDrone-DET2019: The Vision Meets Drone Object Detection in Image Challenge Results.

---

> ### Author Response · Authors · 2024-11-27
> **Response to Reviewer ngxj**
>
> Thank you for your invaluable suggestions to improving our work. We are glad to see your positive evaluation on our contribution to " a new large scale dataset for SSA".  We have included the results from multiple runs in Q1, and one more baseline test in Q2. More analysis are added as replied in Q3.
>
> >**Q1** Multi-run results with mean and variances
>
> To ensure the consistency of our experimental setup, we conducted three runs of the baseline methods for both object detection and object tracking while keeping all parameters and settings unchanged. The mean and standard deviation of the results are presented in Table 2 and Table 4 of the main text and also shown below. From the tables, we observe that the maximum standard deviation for the object detection experiments is below 0.05, while the tracking evaluation metrics have a standard deviation below 0.001. These results confirm the reliability and reproducibility of our experiments.
>
> * Table 1: Performance Comparison of SOTA Models for Space Object Detection
>
> *This table shows the results from multiple runs of each detection method to evaluate consistency and robustness.*
>
> | Model Information   | batch | Mem    | T/epoch | Epochs | Size (MB) | P     | R     | F1    | T/img  |
> |---------------------|-------|--------|---------|--------|-----------|-------|-------|-------|--------|
> | yolov3\_mobilenetv2  | 48    | 34.36G | 5.09s   | 57     | 35.9      | 0.288 ± 0.005 | 0.277 ± 0.023 | 0.282 ± 0.010  | **1.99s** |
> | faster\_rcnn         | 40    | 42.70G | 1.82s   | 24     | 333.8     | 0.347 ± 0.013 | 0.315 ± 0.020 | 0.329 ± 0.009 | 6.42s   |
> | DETR                | 8     | 23.18G | 1.78s   | 209    | 512.2     | 0.236 ± 0.055 | 0.312 ± 0.026 | 0.267 ± 0.043 | 4.60s   |
> | deformable\_detr     | 8     | 38.91G | 4.96s   | 141    | 498.8     | 0.315 ± 0.017 | 0.479 ± 0.024 | 0.380 ± 0.018 | 9.71s   |
> | DINO                | 8     | 34.13G | 6.62s   | 35     | 597.7     | 0.332 ± 0.014 | 0.495 ± 0.092 | 0.394 ± 0.021 | 13.15s  |
> | YOLOv8m             | 48    | 19.20G | 3.81s   | 209    | 52.1      | 0.600 ± 0.017 | 0.435 ± 0.005 | **0.492 ± 0.019** | 3.81s   |
>
>
> * Table 2: Performance Evaluation of Multiple Object Tracking Methods on SpaceSet-100
>
> *This table shows the results from multiple runs of each tracking method to evaluate consistency and robustness.*
>
> | Model                      | Total Target | Predict Target | Real Object | Predict Object | Matches | Misses | FP  | IDs  | MOTA     | Time  |
> |----------------------------|--------------|----------------|-------------|----------------|---------|--------|-----|------|----------|-------|
> | Byte\_iou                  | 4695         | 182            | 56          | 180            | 178     | 4517   | 4   | 143  | 0.0065 ± 0.0002   | 0.06  |
> | Byte\_euclidean            | 4695         | 2111           | 56          | 134            | 2098    | 2597   | 13  | 202  | 0.4012 ± 0.0072   | 0.05  |
> | BoT\_iou                   | 4695         | 182            | 56          | 181            | 178     | 4517   | 4   | 143  | 0.0065 ± 0.0002   | 0.05  |
> | BoT\_euclidean             | 4695         | 2107           | 56          | 150            | 2095    | 2600   | 11  | 183  | 0.4049 ± 0.0078   | 0.05  |
> | BoT\_euclidean\_ecc        | 4695         | 2101           | 56          | 145            | 2092    | 2603   | 9   | 203  | 0.4002 ± 0.0075   | 1.53  |
> | BoT\_euclidean\_orb        | 4695         | 2105           | 56          | 145            | 2095    | 2597   | 9   | 204  | 0.4003 ± 0.0082   | 0.10  |
> | BoT\_euclidean\_sift       | 4695         | 2102           | 56          | 139            | 2092    | 2603   | 10  | 196  | 0.4012 ± 0.0085   | 1.15  |
> | BoT\_euclidean\_sparse     | 4695         | 2106           | 56          | 145            | 2096    | 2599   | 10  | 202  | 0.4017 ± 0.0089   | 0.14  |
> | BoT\_feature\_yolo         | 4695         | 2498           | 56          | 53             | 2486    | 2209   | 12  | 51   | **0.5160 ± 0.0091**   | 0.26  |
> | BoT\_feature\_hog          | 4695         | 499            | 56          | 57             | 495     | 4200   | 3   | 52   | 0.0938 ± 0.0044   | 7.46  |
> | BoT\_feature\_sift         | 4695         | 121            | 56          | 33             | 117     | 4578   | 3   | 4    | 0.0235 ± 0.0010   | 1.42  |
>
> >**Q2** More baseline methods
>
> We carefully reviewed the QueryDet paper [1] and attempted to reproduce its results. On the original VisDrone dataset, our reproduced implementation of QueryDet achieved a precision of 0.503, recall of 0.426, and F1 score of 0.461. However, when applying QueryDet to our dataset using the same parameter settings, the model failed to learn any meaningful information. We tried adjusting key parameters such as the learning rate and batch size, but the training results remained unchanged. A possible reason for this is that the sparse nature of the space dataset significantly affected the training effectiveness of QueryDet.

---

> > ### Author Response · Authors · 2024-11-27
> > **Response to Reviewer ngxj (continued)**
> >
> > >**Q3**  Comprehensive statistical analysis of object scales in dataset
> >
> > We have included the object scale distribution for all RSOs and the scale distributions for sub-classes, namely LEO, MEO, and GEO, in Figure 6 in the revision. From all distributions, most of the RSOs are within 0.5 m to 10 m, and the smallest object is approximately 10 cm. Given the large detection range, detecting such small objects in space is extremely challenging, which highlights the significant contributions of our dataset and our benchmark pipeline.

---

> ### Author Response · Authors · 2024-11-30
>
> Dear Reviewer ngxj,
>
> Thank you once again for taking the time to review our work. We greatly appreciate your feedback and have carefully considered your comments. If you have any remaining concerns after reviewing our responses, we would be happy to discuss them further. Please let us know if you have received our responses and if we have successfully addressed your concerns, as tomorrow is the deadline. Thank you again for your valuable suggestions. Have a great day!

---

> ### Author Response · Authors · 2024-12-02
>
> Dear Reviewer ngxj,
>
> Thank you once again for taking the time to review our work. We greatly appreciate your feedback and have carefully addressed your comments. If you have any remaining concerns after reviewing our responses, we would be happy to discuss them further. Please let us know if you have received our responses and if we have successfully addressed your concerns. Additionally, we would appreciate any insights regarding a potential improvement in the rating score, as tomorrow is the deadline.
>
> Thank you again for your valuable suggestions, and wish you a great day!

---

### Author Response · Authors · 2024-11-27
**Overall Rebuttal by Authors**

We would like to thank all the reviewers (ngxj, EGtv, qNXp, and ajKv) for their valuable suggestions and positive evaluations of our contributions to "a new large-scale dataset for space situational awareness", "the motivation being meaningful and pragmatic in the perspective of addressing limitations in the existing dataset", and "a few merits compared to existing ones". The main concerns mentioned are summarized here, and the reviewers' comments have been addressed one by one. We sincerely hope that all the concerns have been addressed, and we are willing to answer any questions raised during the discussion period. We have attached the revised paper containing new Figures and Tables referenced in our following responses.

>**More dataset analysis**

We have conducted more in-depth analysis (suggested by ngxj and EGtv) beyond the existing SNR, average RMS contrast, and the distribution of observation ranges, including the size distribution of the whole dataset categorized by the classes of LEO, MEO, and GEO in Figure 6. The positional distribution of the observed objects in the image coordinate is also analyzed in Figure 5.

>**More experiments and comparison**

We have included more runs to obtain the mean and standard deviation of the performance (suggested by ngxj). The results show the robust performance of different detection and tracking algorithms. We have also tested one more baseline method [1] as suggested by Reviewer ngxj. The rationale and experimental details (suggested by EGtv) are further explained in the revised paper Section 5.2.

>**Dataset  uniqueness and validity**

We have included more details and explanations of the validation of our dataset. Currently, there are fewer than six datasets in this field, all simulations, as NASA has not made its database public, but our dataset provides the first large-scale realistic space-based image dataset from the photon level, which bridges the gap from simulated to real-world data. Most existing datasets, such as BUAA-SID-share 1.0 [2], SPARK [3], and the Space Target Dataset [4], are generated for satellite pose estimation and space target classification in ideal simulation without noise. These datasets typically focus on capturing targets from close distances and multiple angles to highlight a single target's characteristics. Their approach differs significantly from our dataset, where targets are captured at various distances based on realistic space-based camera specifications and physical models. Compared to the images from the ground-based telescope and the space-based camera in [5], the characteristics of objects and noise distributions highlight the effectiveness of our dataset. We have further explained the realistic characteristics and significant contributions of our dataset for learning representations, as we replied to Reviewer qNXp.

We have carefully proofread our paper and made revisions to avoid any typos and grammatical errors (suggested by EGtv).

[1] Yang, Chenhongyi, et al. "QueryDet: Cascaded sparse query for accelerating high-resolution small object detection." In Proceedings of the IEEE/CVF Conference on computer vision and pattern recognition, pp. 13668-13677. 2022.

[2] Haopeng Zhang, Zhengyi Liu, Zhiguo Jiang, Meng An, and Danpei Zhao. "Buaa-sid1.0 space object image dataset". Spacecraft recovery & remote sensing, 31(4) (2010):65–71.

[3] Musallam, Mohamed Adel, Vincent Gaudilliere, Enjie Ghorbel, Kassem Al Ismaeil, Marcos Damian Perez, Michel Poucet, and Djamila Aouada. "Spacecraft recognition leveraging knowledge of space environment: simulator, dataset, competition design and analysis." In 2021 IEEE International Conference on Image Processing Challenges (ICIPC), pp. 11-15. IEEE, 2021.

[4] Zhang, Zipeng, Chenwei Deng, and Zhiyuan Deng. "A diverse space target dataset with multidebris and realistic on-orbit environment." IEEE Journal of Selected Topics in Applied Earth Observations and Remote Sensing 15 (2022): 9102-9114.

[5] Dignam, Aishling, et al. "In-Space Situational Awareness: Developing Spaceborne Sensors for Detecting, Tracking and Characterising Space Debris." Proc. 2nd NEO and Debris Detection Conference. 2023.

---

### Meta-Review · Area_Chair_MnX4 · 2024-12-23

**Metareview:**

## Summary
The paper introduces SpaceSet, a large-scale realistic space-based image dataset for space situational awareness. It contains 20k images with 673 objects, and includes a pipeline for data curation and annotation. The dataset addresses limitations in existing simulated datasets and provides high-resolution images suitable for advanced Space Situational Awareness methods.

## Strengths
* Proposes the first realistic image dataset at the photon level for space situational awareness, and constructs an object detection and tracking benchmark
* Implements several baseline methods to validate its effectiveness.
* Features high-resolution scenes, more RSOs, and multi-camera images considering various physical factors.
* Enhances applicability to real-world space situational awareness tasks.

## Weaknesses
* The need for detailed information on the dataset, including properties of RSOs, positional distribution of frequently appearing locations, and criterion for image splits.
* The paper's completeness is questioned due to awkward sentences, typos, and grammatical errors.
* The content in A.5 intended to validate the proposed dataset’s effectiveness in real-world data is not clearly conveyed.
* The paper's contribution may not align with the topic of the ICLR conference (Learning Representations), as it benchmarks many existing detection and tracking algorithms on the proposed subset (SpaceSet-100).
* The proposed dataset setup is questioned, with questions about the gap between the proposed synthetic and realistic settings.
* The paper also lacks an available real dataset to evaluate the quality of the synthetic data or the synthetic-to-real generalization ability of an algorithm.

## Conclusions
Two reviewers has not enough expertise to assess the quality of the paper although one of them mentioned that the paper would fit in a AI4Science. The other reviewers have more concerns about the paper although there was not almost no discussion during the rebuttal. Based on the reviews and author’s feedback, the paper should have a more extensive experinents and comparisons before being accepted. For instance, how the size of the slices affects the methods, the qualititive difference between Object detectors, apply SOTA algorithms in OD and tracking.

**Additional Comments On Reviewer Discussion:**

Two reviewers has not enough expertise to assess the quality of the paper although one of them mentioned that the paper would fit in a AI4Science. The other reviewers have more concerns about the paper although there was not almost no discussion during the rebuttal.

---

### Decision · Program_Chairs · 2025-01-22

Reject